# UHPLC-Q-Exactive Orbitrap MS/MS-Based Untargeted Metabolomics and Molecular Networking Reveal the Differential Chemical Constituents of the Bulbs and Flowers of *Fritillaria thunbergii*

**DOI:** 10.3390/molecules27206944

**Published:** 2022-10-16

**Authors:** Xin Li, Pan Wang, Yingpeng Tong, Jie Liu, Guowei Shu

**Affiliations:** 1School of Food and Engineering, Shaanxi University of Science and Technology, Xi’an 710021, China; 2Institute of Traditional Chinese Medicine Industry Innovation of Pan’an, Jinhua 322300, China; 3Institute of Natural Medicine and Health Products, School of Pharmaceutical Sciences, Taizhou University, Taizhou 318000, China; 4Zhejiang Provincial Key Laboratory of Plant Evolutionary Ecology and Conservation, Taizhou University, Taizhou 318000, China

**Keywords:** *Fritillaria thunbergii*, molecular networking, metabolomics, alkaloids, flavonoids

## Abstract

Both the bulbs and flowers of *Fritillaria thunbergii* Miq. (BFT and FFT) are widely applied as expectorants and antitussives in traditional Chinese medicine, but few studies have been conducted to compare the chemical compositions of these plant parts. In this study, 50% methanol extracts of BFT and FFT were analyzed via UHPLC-Q-Exactive Orbitrap MS/MS, and the feasibility of using non-targeted UHPLC-HRMS metabolomics and molecular networking to address the authentication of bulb and flower samples was evaluated. Principal component analysis (PCA), Orthogonal Partial Least Squares-Discriminant Analysis (OPLS-DA), and heat map analysis showed there were dissimilar metabolites in BFT and FFT. As a result, 252 and 107 peaks in positive ion mode and negative mode, respectively, were considered to represent significant difference variables between BFT and FFT. Then, MS/MS-based molecular networking of BFT and FFT was constructed to perform an in-depth characterization of the peaks using different variables. A total of 31 alkaloids with significant differences were annotated in this paper, including seven cis-D/E-vevanine without C20-OH and one trans-D/E-cevanine with C20-OH, thirteen trans-D/E-cevanine without C20-OH, five cevanine N-oxide, and five veratramine. Among the 31 alkaloids, eight alkaloids had higher FFT than BFT contents, while all the flavonoids identified in our work had greater FFT than BFT contents. The influence of different ingredients on the pharmacological activities of BFT and FFT should be investigated in future studies.

## 1. Introduction

*Fritillaria thunbergii* Miq. (*F. thunbergii*) is a perennial herbaceous plant that is widely cultivated in the south-eastern coastal, south-central, and eastern areas of China, mostly in the provinces of Zhejiang, Jiangsu, Anhui, and Hunan. This plant has been grown commercially in China over the last 700 years and is now extensively raised [1]. It grows in partial shade under deciduous shrubs or small trees, where their tall, slender stems form a cluster. In Zhejiang, flowers of *F. thunbergii* usually appear from late February to early April, the stems die in April, and the bulbs were harvested in May and dried for use future [1]. The bulb of *F. thunbergi**i* (BFT), named ”Zhe beimu” in Chinese, is a popularly used herbal medicine in the clinical practice of traditional Chinese medicine (TCM) [1]. Chinese Pharmacopoeia (2020 Edition) reveals that BFT is effective in clearing heat, resolving phlegm, relieving cough, removing toxins, dispelling toxins, and dissolving carbuncles. [2]. There were two different harvest periods for the flower of *F. thunbergii* (FFT): one was before the first bud bloomed, and the other was when the last two buds were blooming. The former is discarded as waste mainly to promote bulb growth, and the latter is a traditional but uncommon Chinese herbal medicine used as a cough and phlegm treatment, as listed in Chinese Pharmacopoeia (1977 Edition). Based on the recording of the Chinese Pharmacopoeia, BFT and FFT have similar clinical uses in the treatment of cough and phlegm in TCM. Therefore, it is interesting and meaningful to explore the similarities and differences between BFT and FFT in terms of their chemical composition and pharmacological activities. Such an analysis is also conducive to the comprehensive development and utilization of FFT.

In clinical studies, BFT has been highly ranked among antitussives, bronchodilators, and expectorants. In modern experimental studies, both micropowders and aqueous extracts of BFT have been found to significantly reduce the frequency of cough and increase the remission period [3], indicating antitussive activity. Isosteroidal alkaloids isolated from other *Fritillaria* plants, including peimine, imperialine, verticinone, ebeiedine, and verticine, have also been identified in BFT and found to exhibit potent antitussive effects [4,5]. Peimine and peiminine have been extracted from FFT via supercritical fluid extraction [6], indicating that FFT may also have antitussive activities, as confirmed by a Chinese research team in 2016 [7]. Indeed, a drug (a tablet of FFT extract) used for the treatment of cough and phlegm was approved by Chinese drug regulatory authorities. Experimental studies have shown that both BFT and FFT have anticancer activities [8]. Studies have confirmed that BFT has anticancer effects on lung, liver, breast, and other cancers, but only one systematic pharmacological study has been conducted on the therapeutic effects of FFT on lung cancer [3,8,9]. It is also believed that *Fritillaria* plants have anti-tumor properties because of containing isosteroidal alkaloids, such as imperialine, verticinone, ebeiedine, verticine, puqietinone, and peimisine [10,11]. BFT extracts exert antiviral effects against the influenza H1N1 virus [12], and the active compounds in BFT extracts function against influenza and influenza-associated diseases, mainly due to containing beta-sitosterol and pelargonidin [13]. BFT water extract was evaluated as a pharmacological candidate for reducing menopausal osteoporosis and menopause-related symptoms, such as fat accumulation [14], which result from the inhibitory effects of peiminine [15]. Therefore, no evidence has been presented in modern studies on pharmacological activities commom to BFT and FFT, other than antitussive and anticancer activities. The similarities and differences in pharmacological activities of BFT and FFT must be attributed to the similarities and differences in their chemical components. It is generally believed that the pharmacological activity of traditional Chinese medicine is determined by both the type of chemical components and the synergistic effects of components [16,17,18]. Therefore, clarifying the similarities between the chemical composition of BFT and FFT can provide an important reference for subsequentr pharmacodynamic evaluations.

To the best of our knowledge, only 15 and 8 chemical structures of compounds isolated from BFT and FFT via column chromatography have been elucidated using modern instrumental analysis (IR, ESI-MS, HR-ESI-MS, 1D and/or 2D NMR) [3,19]. Appendix A provides detailed information on the chemical constituents of *F.*
*thunbergi**i.*, showing no compounds common to BFT and FFT. LC-LTQ-Orbitrap MS^n^ was applied to identify the alkaloids and flavonoids in BFT and FFT [20]. Twenty-eight and thirty alkaloids, were identified in BFT and FFT, respectively, with 18 alkaloids in common. Sixteen flavonoids were identified in FFT, but no flavonoids were detected in BFT. Only chemical differences between BFT and FFT were initially compared in this study. Only seven alkaloids were identified after disregarding undetectable isomers. Therefore, more studies are needed to fully explore the chemical ingredients of BFT and FFT, especially the similarities and differences in chemical classes and contents thereof.

Metabolomics seeks to identify variances in global metabolite fingerprints, offering considerable advantages for detecting chemical components that differ significantly from one another [21,22]. A metabolomic study on *F. thunbergii* was conducted based on the results of a gas chromatopraphy-mass spectrometry (GC-MS) analysis, and five compounds, including tryptophan, gentiobiose, xylose, phenylalanine, and 3-oxovaleric acid, were determined to play an important role in distinguishing bulbs from flowers [23]. However, GC-MS can mainly detect low-polarity components and cannot analyze the main active components of BFT and FFT such as alkaloids and flavonoids. Therefore, futher metabolomic analyses need to be carried out in more detail, based on other chemical analysis methods such as UHPLC-MS^n^ and NMR.

Currently, ultra-high-performance liquid chromatography with high-resolution mass spectrometry (UHPLC-Q-Orbitrap)-based metabolomics has been widely used to identify the significant different chemical constituents in various herb samples [24,25]. A challenge in metabolomics analysis is the qualitative analysis of large quantities of MS data. Molecular networking (MN) is a useful tool for categorizing ingredients with similar molecular structures into clusters via MS/MS-based metabolomic experiments. Comparison against the mass spectrometry datasets in the Global Natural Products Social Molecular Networking (GNPS) has proven to be a powerful method in the chemical identification of metabolomics [26,27].

In summary, few studies have been performed on the chemical differences between BFT and FFT, which constitute the main obstacle for activity evaluations, bioactivity substance screening, and product development of FFT in the future. Untargeted metabolomics and molecular networking based on UHPLC-Q-Orbitrap MS/MS were used in the present study to overcome this challenge.

## 2. Results and Discussion

### 2.1. Metabolomic Profiling of BFT and FFT

The typical base peak chromatograms (BPC) of BFT and FFT extracts obtained via UPLC-Q-Orbitrap-MS/MS in positive ion mode (ESI+) and negative ion mode (ESI−) mode were shown in Figure 1. The metabolites from FFT presented higher sensitivity in ESI− mode than those from BFT (Figure 1a,b). However, visual inspection of the BPCs in ESI− mode alone was not sufficient to determine metabolite differences between BFT and FFT because the same metabolites, such as alkaloids in BFT, exhibited higher sensitivity in ESI+ mode than in ESI− mode (Figure 1a,c), which is consistent with the results of previous studies [28,29,30]. Therefore, MS data from both ESI+ and ESIESI− modes were further compared and annotated. These data were processed using MZmine 2.5.3, and a total of 863 and 349 precursor ion peaks were extracted in the ESI+ and ESI− modes, respectively, for further analysis. The data matrices generated based on the intensity of the precursor ions peaks in ESI+ and ESI− modes were imported into SIMCA 14.0 for PCA and OPLS-DA.

The PCA score scatter plot (Figure 2a,d) revealed a clear separation between the chemical profiles of BFT and FFT. However, the variation in the chemical profile of FFT was more prominent than that of BFT, indicating that the harvest time had a strong influence on the FFT constituents (Figure 2a). In the market for Chinese herbal medicines, the BFT diameter is one of the most important indicators of the BFT grade, that is, the larger the BFT diameter is, the higher the quality is considered to be. Figure 2d shows different results in the metabolomic profiles of BFTs with different diameters. The metabolomic profiles of BFT could be further categorized into two subgroups with diameters below and above 2 cm. Therefore, a diameter of 2 cm may be a key index for assessing BFT quality.

To better observe the chemical variation between BFT and FFT, OPLS-DA analysis was performed, and the score plot model was shown in Figure 2b,e. The OPLS-DA models possessed great fitting and prediction abilities, as reflected in the evaluation parameters: R2Xcum = 0.774, R2Ycum = 0.999, and Q2cum = 0.972 for ESI+ mode and R2Xcum = 0.918, R2Ycum = 0.999, and Q2cum = 0.987 for ESI– mode. In addition, a 200-times permutation test was performed to determine whether the two new OPLS-DA models experienced over-fitting. In Figure 2c,f, all the permuted R2, and Q2 values to the left were lower than the original points to the right, strongly indicating that the OPLS-DA models are valid.

The variable importance on projection (VIP) was used to evaluate the discriminatory ability of each compound in BFT and FFT. A compound with a VIP score value of greater than one was considered to be a significant variable to discriminate the extracts from BFT and FFT. In the present work, there were 491 and 216 peaks, respectively, in the PI and NI models with VIP values greater than one, further confirming the significant differences in compounds between BFT and FFT. An ANOVA analysis was also conducted to screen for significant variables, and the cut-off level of *p*-value and |log2(FC)| was set at 0.01 and 2. After screening with VIP for the *p*-value and |log2(FC)|, 252 and 107 peaks in the ESI+ and ESI− models, respectively, were found, indicating significant differences in variables between BFT and FFT.

### 2.2. Molecular Networking

UHPLC-MS/MS-based molecular networking is a useful tool for identifying chemical structures in complexes. This analysis method consists of comparing MS/MS spectra to cluster compounds with similar structures. In the present study, molecular networking was performed by using MS data obtained in the ESI+ and ESI− modes following a feature-based workflow. Figure 3 shows these modes visualized using Cytoscape. Molecular networking was performed in ESI+ mode using 68 clusters (node ≥ 2) and 242 single nodes. The number of clusters (node ≥ 2) and single nodes in molecular networking under ESI− mode totaled 18 and 65, respectively.

The intensities of compounds related to individual nodes in BFT and FFT were compared, and the results are shown in different colors (see Figure 3). The nodes shown in red, green, and gray represent the contents of compounds for which there were significant increases, decreases, or no difference between BFT and FFT (*p* < 0.01). These results- showed the number of node-related compounds with significantly higher contents in FFT compared to BFT was higher than the number of node-related compounds with a significantly lower content in FFT compared to BFT (Figure 3C). Notably, only seven of the nodes obtained in negative mode had significantly higher content in BFT than in FFT because some compounds obtained in ESI− mode could not be detected. However, there were still more nodes with higher contents in FFT than in BFT that were obtained in ESI− mode than in ESI+ mode, indicating that the chemical constituents in FFT were more abundant than in BFT.

We annotate the chemical structures of nodes with significantly different contents using MolNetEnhancer, a workflow that emloys ClassFire to combines the outputs from molecular networking, MS2LDA, in silico annotation tools (such as Network Annotation Propagation or DEREPLICATOR), and automated chemical classification to provide a comprehensive chemical overview of metabolomics data whileelucidating the structural details of individual fragmentation spectra [31].

Figure 3 shows that a total of 12 chemical classes were annotated in these two molecular networks, including steroidal saponins (a); linoleic acids (b); amino acids, peptides, and analogs (c); 1-acyl-sn-glycero-3-phosphocholines (d); flavonoids (e); hydroxycinnamic acids (f); depsipeptides (g); aryl phosphotriesters (h); carbohydrates (i); purine nucleosides (j); 1,2-aminoalcohols (k); and fatty acids (l).

### 2.3. Annotated Nodes in Molecular Networking

The characteristic features of the mass spectra of steroidal alkaloids and flavonoids have been thoroughly studied. In this study, the chemical structures of steroidal alkaloids and flavonoids were identified by using molecular networking analysis conjunction with previously reported characteristic features. Other components with significant differences between BFT and FFT were mainly annotated according to the search results of the GNPS database.

#### 2.3.1. Construction of an In-House Database of Isolated Ingredients from Fritillaria Plants

Although several electronic databases for Chinese herbs have been constructed, including TCMSP and TCMIP, the information on components collected in these databases is not comprehensive. In the present study, we constructed an in-house database of ingredients isolated from Fritillaria plants, as shown in Appendix A [1,3,32,33]. In this database, a total of 298 compounds containing 127 alkaloids, 6 fatty acids, 9 flavonoids, 6 phenylpropanoids, 39 steroidal saponins, 51 terpenoids, and 60 other compounds were collected from references. The molecular formulas for the aforementioned compounds were extracted from references or public databases such as PubChem and ChemSpider. We used the theoretically accurate masses of the different atoms in the chemical formulas (H:1.00782503207 Da, C: 12 Da, O: 15.99491461956 Da, N: 14.0030740048 Da, S: 31.972071 Da and cl:34.9689Da), we calculated the theoretically accurate masses of six adduct forms of compounds ([M+H]^+^, [M+Na]^+^, [2M+H]^+^, [M−H]^−^, [M+HCOO]^−^, [2M−H]^−^).

The theoretically accurate masses of six different adducts of each compound in the in-house database—were used to quickly determine a total of 132 and 27 experimental ions in the PI and NE model with mass errors within 10 ppm using a self-built R code. The above mentioned results helped to further refine the structural types of the compounds represented by each cluster in the mass spectrometry molecular network. For example, the clusters in Figure 3a were all annotated as steroidal saponins, whereas those shown in Figure 3A(a1) were further annotated as steroidal alkaloids to match the results to be those in the in-house database and confirm previously reported MS/MS spectra of steroidal alkaloids.

#### 2.3.2. Steroidal Alkaloids

Steroidal alkaloids are the most important active ingredients in BFT and FFT, and several previous studies were conducted to identify steroidal alkaloids via LC-MS/MS [28,29,30,34,35,36,37,38,39,40]. Steroidal alkaloids in the *Fritillaria* plant can be divided into the following four subclasses: cevanines (cis-D/E-cevanines, trans-D/E-cevanines, and cevanine N-oxide), jervines, veratramines, and verazines. These alkaloids can be distinguished by combining key diagnostic fragment ions and the relative abundance and amount of major fragment ions in ESI+ mode [34]. In general, jervine-, veratramine-, and verazine-type steroidal alkaloids have strong ion peaks at *m/z* 114.09, 128.11, and 95.09, respectively. cevanine-type steroidal alkaloids have dominant ion peaks at [M+H]^+^ or [M+H-H_2_O]^+^, depending on whether there is an OH group at the C20 position. cevanine-type steroidal alkaloids can be further divided into five types, including cis-D/E-cevanine with C20-OH, trans-D/E-cevanine with C20-OH, cis-D/E-cevanine without C20-OH, trans-D/E-cevanine without C20-OH, and cevanine with N-oxide. Cevanine with N-oxide has dominant ion peaks at *m/z* 112.1133, while cis-D/E-cevanine with C20-OH and cis-D/E-cevanine without C20-OH have dominant ion peaks at *m/z* 138.13 and 98.09 [29].

Interestedly, molecular networking can group the same types of steroidal alkaloids into one sub-cluster (Figure 4), further confirming MN as a useful tool to identify compounds. In the present work, only cevanine- (cis-D/E-cevanine without C20-OH, trans-D/E-cevanine with C20-OH, trans-D/E-Cevanine without C20-OH, and cevanine N-oxide) and veratramine-type of steroidal alkaloids were identified. In addition to the compounds matched with the in-house database, other putatively identified steroidal alkaloids were constructed following the method of Liu [29] and listed in Appendix A in the chemical name format. As a result, most of the nodes were annotated, including twenty-two cis-D/E-cevanine without C20-OH, ten trans-D/E-cevanine with C20-OH, twenty-one trans-D/E-cevanine without C20-OH, nine cevanine N-oxide, and nine veratramine.

#### 2.3.3. Flavonoids

The flavonoid nodes that were annotated by MolNetEnhancer were then detected in the molecular networks of both the ESI+ and ESI− models. The sub-molecular network of flavonoids was shown in Figure 5. In previous studies, negative ionization has been demonstrated to provide the highest sensitivity for flavonoids [41]. Therefore, the flavonoids were mainly identified based on the characteristic fragment ions collected in ESI− mode. UPLC-MS/MS is frequently used to identify flavonoids [42,43,44]. Flavonoids were identified using the following steps in this study.Firstly, the molecular formula of each precursor ion was calculated using Xcalibur software. Secondly, the aglycones of the flavonoids were proposed according to previously reported characteristic fragment ions (Appendix A). Thirdly, the classes and numbers of sugar residues were calculated from the losses of neutral sugar residues (162, 146, and 132 amu for hexosides, deoxyhexosides, and pentosides, respectively) between precursor ions and flavonoid aglycones. Lastly, the link positions between sugar–aglycone, and sugar–sugar were inferred. In general, the intensity of [Y0-H]^−^ was sometimes higher than that of [Y0]^−^, indicating that the glycosylation position was at either 3-OH or 7-OH. It was also found that the [Y0]^−^ ion was always the base peak, whereas the [Y1]^−^ ion was more abundant in the 1→2 linkage than in 1→6 linked glycosides for O-diglycosides. Most of the time, the negative-ion MS/MS spectra of 1→6 glycosides did not indicate the presence of a significant number of [Y1]^−^ ions.

Figure 5 and Appendix A, show the 16 identified flavonoids, including quercetin, luteolin, 2 taxifolin isomers, 1 kaempferol glycoside, 7 quercetin glycosides, and 3 isorhamnetin glycosides. In addition to quercetin 3-rutinoside-7-glucoside, six other quercetin glycosides were included in cluster (a) (see Figure 5), and three isorhamnetin glycosides were included in cluster (b), demonstrating that compounds with similar structures could be grouped into the same clusters via molecular networking. Therefore, MN was found to be a useful tool for identifying compounds via LC-MS/MS analysis. In the present work, nodes related to some unusual adduct ions, such as [M−H+HCOONa]^−^ and [M+NaSO4]^−^, were also included in Figure 5a,b, showing the same RT and similar fragmentation patterns to the [M−H]^−^ ions of flavonoids with high contents in FFT, as shown in Appendix A. These kinds of ions have been previously detected in oligosaccharides [45]. At least three flavonoid aglycones are presented in Figure 5b1, but these aglycones have the same RT as the nodes are shown in Figure 5b2. The nodes in Figure 5b2 represent flavonoid glycosides with shorter RTs than those of flavonoid aglycones. Therefore, the nodes in Figure 5b1 were annotated as the product ions of flavonoid glycosides in FFT.

### 2.4. The Differences and Similarities of Chemical Ingredients between BFT and FFT

In the present study, the ingredients of steroidal alkaloids and flavonoids were identified by combining MN analysis with the corresponding MS/MS spectra. Other ingredients were identified by comparison of the MS/MS spectra with those of compounds in the GNPS library, and are listed in Appendix A. As shown in Figure 6 shows significant differences for alkaloids, flavonoids, and other ingredients were distinguishable between BFT and FFT, respectively. A total of 71 alkaloids were annotated in this paper, of which 31 exhibited significant differences between BFT and FFT(Figure 6a), including seven cis-D/E-Cevanine alkaloids without C20-OH and one trans-D/E-cevanine with C20-OH, thirteen trans-D/E-cevanine alkaloids without C20-OH, five cevanine N-oxide alkaloids, and five veratramine. Among the cevanine alkaloids, which accounted for over 80% of all available steroidal alkaloids in Fritillaria plants, trans-D/E-cevanine with C20-OH alkaloids had the highest contents in BFT, where two trans-D/E-cevanine alkaloids with C20-OH, named peimine and peininine, were selected as the quality markers in Chinese Pharmacopoeia (2020 Edition). While only one trans-D/E-cevanine alkaloid with C20-OH exhibited significantly different intensities between BFT and FFT, there were no significant differences for the other nine trans-D/E-cevanine alkaloids with C20-OH, including peimine and peininine. peimine and peininine are considered the main active ingredients for cough treatments, indicating that FFT extract is also useful for cough treatment. Most trans-D/E-cevanine alkaloids without C20-OH and veratramine alkaloids presented significant differences between BFT and FFT, accounting for 61.9% and 55.6% of the identified alkaloids, respectively. The contents of all trans-D/E-cevanine alkaloids without C20-OH and veratramine alkaloids with significant differences between BFT and FFT were all higher in BFT than in FFT. Among the 22 identified cis-D/E-cevanine alkaloids without C20-OH, those with significant differences made up approximately 31.8% of the total, and most had higher contents in FFT than in BFT. Approximately 55.6% of Cevanine N-oxide alkaloids differed significantly in content between BFT and FFT, and highest contents in BFT and FFT corresponded to three and two alkaloids, respectively.

A total of 10 flavonoids differed significantly between BFT and FFT, accounting for 62.5% of the total number of flavonoids, and the contents of these 10 components were all significantly higher in FFT than in BFT. Quercetin glycosides, except for taxifolin, were flavonoids that did not exhibit significant differences between FFT and BFT.

Other identified ingredients with significant differences between BFT and FFT belonged to 1-acyl-sn-glycero-3-phosphocholines, depsipeptides, hydroxycinnamic acids, linoleic acids and amino acids, peptides, and related analogs. The contents of 1-acyl-sn-glycero-3-phosphocholines and amino acids, peptides, and analogs in BFT were all significantly higher in BFT than in FFT.

## 3. Materials and Methods

### 3.1. Sample Preparation

BFT and FFT were collected from Xinwo Street, Pan’an, Zhejiang Province, China. The samples were authenticated by Dr. Jie Liu and deposited at Taizhou University. The FFT samples were harvested at different times of the year in 2019. The BFT samples were harvested at the same times of year in 2019 but had different diameters. Detailed information on the samples used in this study is presented in Table 1.

The fresh BFT and FFT were thoroughly dried in an oven at 60 °C and then pulverized into powder using a sample mill. Approximately 1 g of sample powder, individually sieved through a 50-mesh (0.25 mm, I.D.) grid, was accurately weighed and put in a 100-mL conical flask with a stopper. Then, 40 mL of a 50% methanol aqueous solution was added to the flask and the resulting mixture was ultrasonically extracted for 1 h in a 60 °C water bath. Next, 50% methanol was added to the flask to obtain a prescribed weight, the lid was screwed on, and the conical flask was shaken evenly. Prior to analysis, the sample solution was filtered through a 0.22 μm filter membrane.

### 3.2. UHPLC-Q-Orbitrap-MS/MS Analysis

UHPLC-Q-Orbitrap-MS/MS analysis was conducted using an Ultimate RS3000 UHPLC (Thermo Fisher Scientific, Carlsbad, CA, USA) connected to a Q-Exactive™ mass spectrometer (Thermo Scientific, Sunnyvale, CA, USA) with heated electrospray ionization (HESI-II; Thermo Fisher Scientific). The extracts were chromatographically separated on an Hypersil GOLD^TM^ device (2.1 × 100 mm, 3.0 μm, Thermo Scientific) using a mobile phase consisting of 0.5% formic acid/water (A) and acetonitrile (B). Gradient elution was performed with the following protocol: 0–10 min, 5% B; 10–20 min, 5–40% B; 20–45 min, 40–90% B; and 45–50 min, 90% B. The column was maintained at 30 °C, the flow rate was 0.3 mL/min and the injection volume was 2 μL.

The mass spectrum data was acquired in both positive and negative ion mode through full MS and higher energy collisional dissociation (HCD) data-dependent MS/MS by the UHPLC-Q-Orbitrap-MS/MS equipped with a control software (Xcalibur, version 4.2.27). The scan parameters were set as follows: scan range, 100 to 1500 *m/z*; resolution, 70000; AGC target, 3 × 10^6^; maximum inject time, 100 ms. The following parameters were used for the heated electrospray ionization source (HESI source): spray voltage, 3.0 kV for ESI− and 3.5 kV for ESI+; sheath gas flow rate, 40 arb; auxiliary gas flow rate, 10 arb; sweep gas flow rate, 0 arb; capillary temperature, 320 °C; s-lens RF level, 50; and auxiliary gas heater temperature, 350 °C. A data-dependent scan (dd-MS2) was applied to acquire high-quality MS2 data. The top 5 most intense precursors were automatically selected for MS/MS fragmentation by HCD. The parameters of dd-MS2 were as follows: resolution, 17,500; AGC target, 1 × 10^5^; Maximum IT, 50 ms; Loop count, 5; isolation window, 4 *m/z*. The Normalized Collision Energy (NCE) was set at 20, 40, and 60 V.

### 3.3. Data Processing via MZmine 2.5.3

The complete GNPS package was downloaded (https://ccms-ucsd.github.io/GNPSDocumentation/fileconversion/, accessed on 1 July 2021), the raw mass data acquired by UPLC-Q-Orbitrap MS/MS analysis were converted into an mzXML format.

Then, the data were imported into MZmine 2.53 and further processed according to our previous study with minor modifications. Briefly, the noise levels for mass detection of MS1 and MS2 were set to 1 × 10^5^ and 1 × 10^3^, respectively, for mass detection. An *m/z* tolerance of 10 ppm was applied to chromatograms with a minimum peak height of 1 × 10^5^ and an *m/z* tolerance of 0.005 Da. A wavelet (ADAP) algorithm was employed to deconvolve the chromatograms using a peak height of 5 × 10^5^. Peaks were grouped according to their isotope composition with an *m/z* tolerance of 0.005 Da or 10 ppm, and an Rt (retention time) tolerance of 0.5 min. Peaks were filtered by having at least three peaks in a row. The processed data were exported as CSV and MGF files for further processing.

### 3.4. Chemometric Analysis

Simca 14.1 software (Umetrics, Ume, Sweden) was used to perform PCA and OPLS-DA on data exported from MZmine 2.5.3 in a CSV file format. The R2 and Q2 coefficients are commonly used to evaluate the quality and reliability of these models, where values close to 1.0 indicate that a model has an excellent fitness and predictive capacity. The variable importance in projection (VIP) is a useful and simple tool for assessing the importance of variables in OPLS models. The *p*-value in ANOVA was also used to screen compounds with significant differences in content between BFT and FFT. In this study, cutoff values of VIP > 1 and *p* < 0.01 were applied to filter the significant compounds in the flowers and bulbs of *F. thunbergii*.

### 3.5. An Integrated Strategy for Chemical Identification

We proposed an integrated interpretation strategy for in-depth profiling of significant compounds in the flowers and bulbs of *F. thunbergii*, for which the steps are given below.

This strategy began with the collection of information on identified compounds published in the literature to build an in-house database. Information on compound names, molecular formulas, and adduct ions ([M+H]^+^, [M+Na]^+^, [2M+H]^+^, [M−H]^−^, [M+HCOO]^−^, [2M−H]^−^) were collected.

Secondly, the GNPS online platform was used to generate a molecular network from the MS/MS data processed using MZmine 2.5.3. Molecular networks were constructed with a cosine score above 0.6 and a precursor ion mass tolerance of 0.0075 Da, with at least five matched fragment ions. Cytoscape 3.9.1 was used for network visualization.

Thirdly, nodes corresponding to experimental ions with mass errors within 10 ppm compared to the theoretical *m/z* of adduct ions in the in-house database were also recorded. The structures of these matched nodes were then confirmed by comparing the MS/MS data in our study to the fragment ions of these compounds recorded in the references.

Lastly, the compounds that remained unmatched with the in-house database were also identified with the help of GNPS.

## 4. Conclusions

A molecular networking analysis based on UHPLC-Q-Orbitrap MS/MS was used to identify known metabolites through spectral library matching, and putative metabolites in *Fritillaria* plants that have not been previously described were annotated. The untargeted metabolomic analysis revealed significant differences in the chemical ingredients of BFT and FFT, except for the main active ingredients that had high contents in BFT, as suggested by previous reports. Flavonoids were the main ingredients with high contents in FFT, and some of these flavonoids were not detected in BFT. Future studies should focus on using more computational tools to discover metabolites and developing automated approaches to investigate complex metabolomes. More experiments should be carried out to investigate why differences in ingredients affect the pharmacological activity.

## Figures and Tables

**Figure 1 molecules-27-06944-f001:**
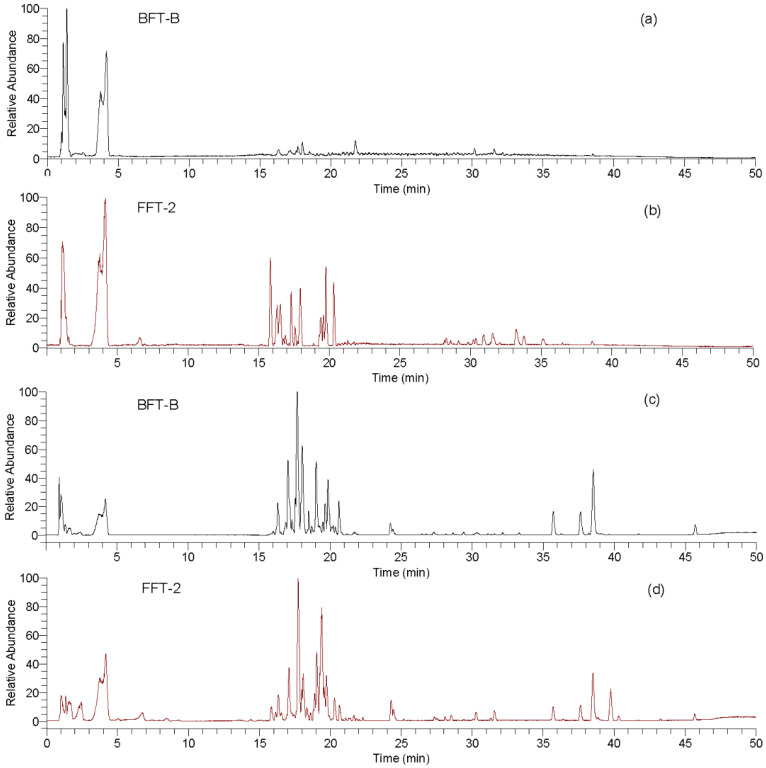
The total ion chromatograms of flower (FFT-2) and bulb (BFT-B) of *F. thunbergii* under negative (**a**,**b**) and positive (**c**,**d**) ion mode.

**Figure 2 molecules-27-06944-f002:**
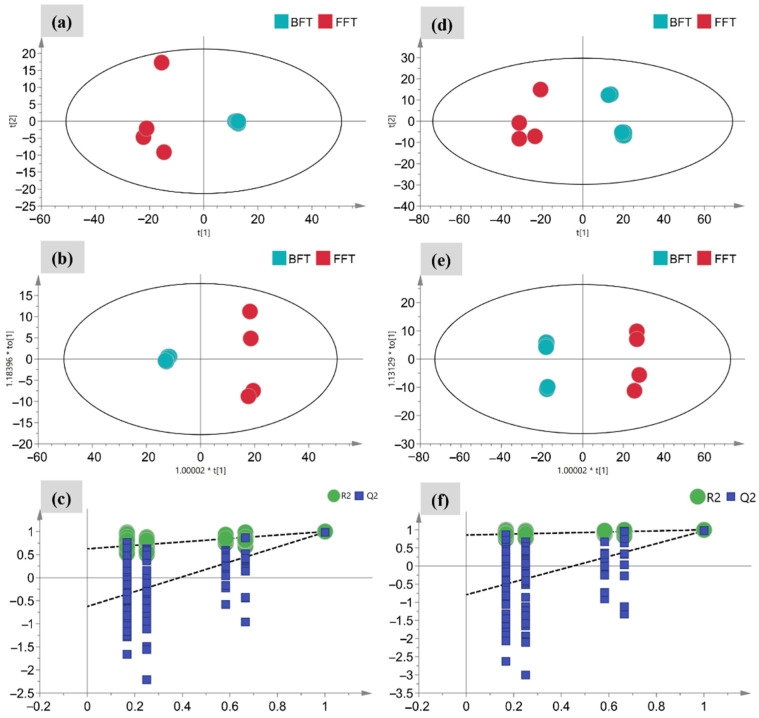
Multivariate statistical analyses of the flower and bulb of *F. thunbergii* under negative (**a**–**c**) and positive (**d**–**f**) ion modes: PCA score plot (**a**,**d**); OPLS-DA score plot (**b**,**e**); cross-validation plot of the OPLS-DA model with 200 permutation tests (**c**,**f**).

**Figure 3 molecules-27-06944-f003:**
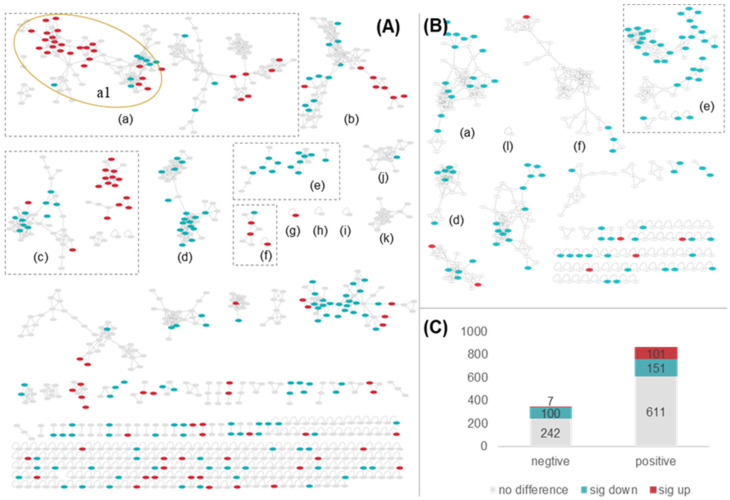
Feature-based molecular networking of chemical constituents from BFT and FFT in PI (**A**) and NI (**B**) model. The clusters marked with different lowercase letters represent the chemical classes annotated by MolNetEnhancer (steroidal saponins (a), steroidal alkaloids (a1), lineolic acids (b), amino acids, peptides, and analogs (c), 1-acyl-sn-glycero-3-phosphocholines (d), flavonoids (e), hydroxycinnamic acids (f), depsipeptides (g), aryl phosphotriesters (h), carbohydrates (i), purine nucleosides (j), 1,2-aminoalcohols (k) and fatty acids (l)). The nodes with red, green, and grey colors represent the contents of their corresponding compound between BFT and FFT are significantly up, significantly down, and have no difference (*p* < 0.01), respectively. The numbers of nodes with different color in negative and positive model were summarized (**C**).

**Figure 4 molecules-27-06944-f004:**
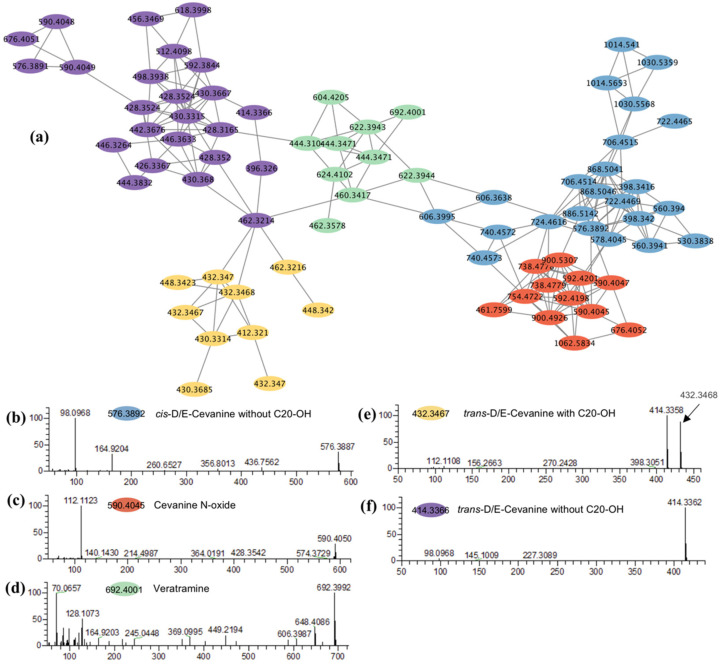
Molecular networking of steroidal alkaloids (**a**), The typical MS/MS spectra of *cis*-D/E-cevanine without C20-OH (**b**), cevanine N-oxide (**c**), veratramine (**d**), *trans*-D/E-cevanine with C20-OH (**e**) and *trans*-D/E-cevanine without C20-OH (**f**).

**Figure 5 molecules-27-06944-f005:**
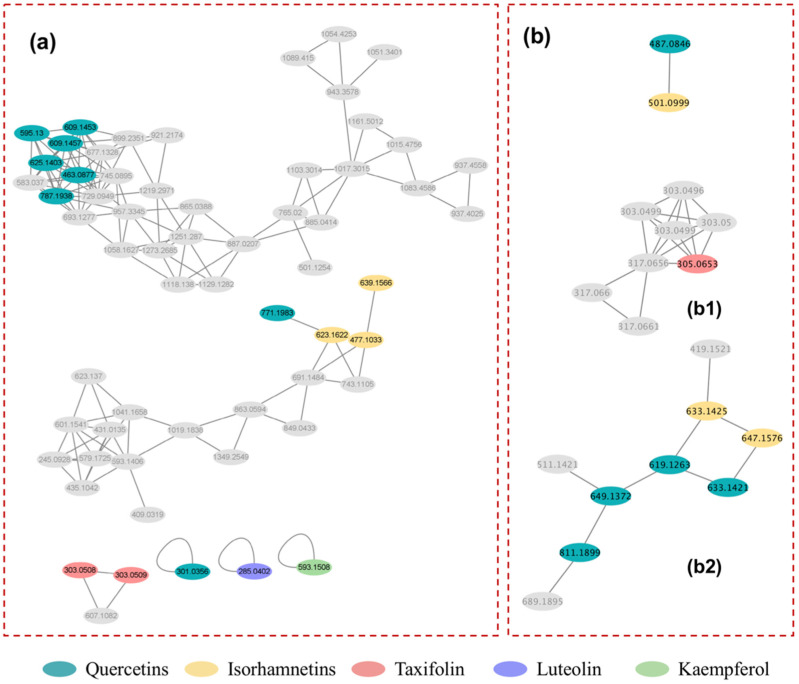
Molecular networking of flavonoids in negative (**a**) and positive model (**b**). The sub-molecular networking of part flavonoid aglycones (**b1**) and their flavonoid glycosides (**b2**).The nodes with cyan, yellow, red, purple and green color represent the compounds with flavonoid aglycones of quercetin, isorhamnetin, taxifolin, luteolin and kaempferol, respectively.

**Figure 6 molecules-27-06944-f006:**
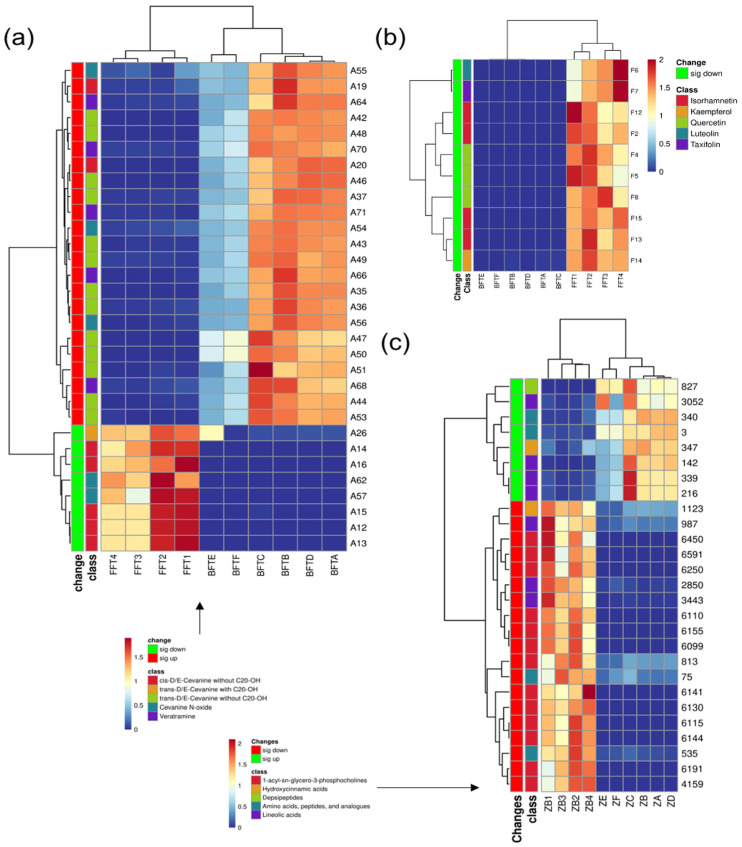
Heatmap of ingredients with significant differences between BFT and FFT ((**a**): steroidal alkaloids; (**b**): flavonoids; (**c**): other ingredients).

**Table 1 molecules-27-06944-t001:** Detailed information for the samples used in this study.

No.	Medicinal Parts	Materials Information
FFT-1	Flower	February 20th
FFT-2	Flower	March 1st
FFT-3	Flower	March 10th
FFT-4	Flower	March 20th
BFT-A	Bulb	Diameter ≤ 1.0 cm
BFT-B	Bulb	1.0 cm < Diameter ≤ 1.5 cm
BFT-C	Bulb	1.5 cm < Diameter ≤ 2.0 cm
BFT-D	Bulb	2.0 cm < Diameter ≤ 2.5 cm
BFT-E	Bulb	2.5 cm < Diameter ≤ 3.0 cm

## Data Availability

Data will be provided upon request.

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
