# Peer review of "UHPLC-Q-Exactive Orbitrap MS/MS-Based Untargeted Metabolomics and Molecular Networking Reveal the Differential Chemical Constituents of the Bulbs and Flowers of Fritillaria thunbergii"

_molecules, 2022, doi:10.3390/molecules27206944_

Round 1

Reviewer 1 Report

The authors of the manuscript “UHPLC-Q-Exactive Orbitrap MS/MS-based untargeted metabolomics and molecular networking reveal the differential chemical constituents of the bulbs and flowers of Fritillaria thunbergii” present an articulated work about the comparison of chemical composition of bulbs and flowers of Fritillaria thunbergii, a popular herbal medicine in the clinical practice of traditional Chinese medicine (TCM) using UHPLC-MS/MS-based molecular networking.

The manuscript is clearly written and the results are of interest but, it contains some flawn to be improved before acceptance.

The “Fritillaria thunbergii” is not always spelled correctly. Please correct.

In Figure S1, the labels do no match those in the text.

Page 4, line 152: “(c-f) should be substituted with (d-f)

Page 3 line 145: is “Figure 2d” a mistake?

Page 6 line 230-231: “the clusters in Figure S1-a were all annotated as steroidal saponins, while those shown in Figure S2-a1 were further annotated as steroidal alkaloids”. Please correct with the correct test.

Page 8, line 290: is “Fig S1-a and b” a mistake?

Author Response

Point 1: The “Fritillaria thunbergii” is not always spelled correctly. Please correct.

Response 1: Thanks for your careful checks. We are sorry for our spelling mistakes. We have carefully revised the words of “Fritillaria thunbergii” and marked in red in our manuscript. 

Point 2: In Figure S1, the labels do no match those in the text.

Response 2: Thanks for your careful checks. The labels of ZBM-B and BMH-2 should be changed to BFT-B and FFT-2, respectively. We have revised in our manuscript and marked in red. According to reviewers’ suggestions, the figures in supplementary data could be put into our text, therefore we rearranged our figures in the text and the Figure S1 was changed to Figure 1.

Point 3: Page 4, line 152: “(c-f) should be substituted with (d-f)

Response 3: Thanks for your careful checks. We have revised in our manuscript and marked in red.

Point 4: Page 3 line 145: is “Figure 2d” a mistake?

Response 4: Thanks for you careful checks. After the figure is rearranged, it does not need to be modified here.

Point 5: Page 6 line 230-231: “the clusters in Figure S1-a were all annotated as steroidal saponins, while those shown in Figure S2-a1 were further annotated as steroidal alkaloids”. Please correct with the correct test.

Response 5: Thanks for you careful checks. There are two main errors in this area. One is that a1 has forgotten to be marked in the figure. The other is that Figure S1-a should be changed to Figure S2-a.  Because of the figure is rearranged,  Figure S1-a had changed to Figure 3a.  We reprocessed the data, and the results are shown in attachment.

Point 6: Page 8, line 290: is “Fig S1-a and b” a mistake?

Response 6: Thanks for you careful checks. Fig S1-a and b should be changed to Figure S3-a and b, Because of the figure is rearranged,  Figure S1-a and b had changed to Figure 5a and 5b.

Very thanks again for your help.

Reviewer 2 Report

The manuscript conducted a comparative study of the chemical composition of bulbs and flowers of an important Chinese herb Fritillaria thunbergii, which has some significance for further introducing the function of Chinese medicine. The paper should be published after a mini revision.

The authors of the manuscript writing should pay attention to tense and basic spaces in text. it is best to ask an English expert to improve the language; some mini errors has been shown in the PDF file.

Figure S1-3 should change as Figure 1-3, they should be arranged in the text.

Author Response

Point 1: The authors of the manuscript writing should pay attention to tense and basic spaces in text. it is best to ask an English expert to improve the language; some mini errors has been shown in the PDF file.

Response 1: Thanks for your careful checks and correction. We have revised the paper again.

Point 2: Figure S1-3 should change as Figure 1-3, they should be arranged in the text.

Response 2: Your good advice was very much appreciated. We arranged all figures in paper.

Reviewer 3 Report

The research finding supports the basic scientific conclusion that storage organs have more compounds than flowers. Flowers have specific pupose of fertilization and the chemicals present are for attraction/fertilization.

This is a comparative and qualitative study of two plant parts used in the system. A quantitative marker would have added a more conclusive conclusion. The manuscript is well-written and acceptable.

Author Response

Point 1: This is a comparative and qualitative study of two plant parts used in the system. A quantitative marker would have added a more conclusive conclusion. The manuscript is well-written and acceptable.

Response 1: Very thanks for your recommendation and suggestion. This paper we aim to assesses the feasibility of using non-targeted UHPLC-HRMS metabolomics and molecular networking to address the authentication of bulb and flower samples which from same species. So we focus on the qualitative analysis of compounds, and we will  take your advice in the next research working on different species.